# Isochronic development of cortical synapses in primates and mice

Gregg Wildenberg[1,2] ✉, Hanyu Li[1,2], Vandana Sampathkumar[1,2], Anastasia Sorokina[1,2] & Narayanan Kasthuri [1,2] ✉

The neotenous, or delayed, development of primate neurons, particularly human ones, is thought to underlie primate-specific abilities like cognition. We tested whether synaptic development follows suit—would synapses, in absolute time, develop slower in longer-lived, highly cognitive species like non-human primates than in shorter-lived species with less human-like cognitive abilities, e.g., the mouse? Instead, we find that excitatory and inhibitory synapses in the male *Mus musculus* (mouse) and *Rhesus macaque* (primate) cortex form at similar rates, at similar times after birth. Primate excitatory and inhibitory synapses and mouse excitatory synapses also prune in such an isochronic fashion. Mouse inhibitory synapses are the lone exception, which are not pruned and instead continuously added throughout life. The monotony of synaptic development clocks across species with disparate lifespans, experiences, and cognitive abilities argues that such programs are likely orchestrated by genetic events rather than experience.

The extraordinary capabilities of human brains, and primate brains broadly, are thought to emerge from excessively slow development—termed 'neoteny'[1–3]. Evidence for neoteny abounds. Relative to other species, for example, rodents, embryonic primate and human brains produce brain cells for longer periods of time[4,5], reach full size later[6], express developmental genes later[3,7], and take longer to establish basic behaviors like walking[8]. Developmental learning also appears neotenous—post-natal critical periods, periods when experiences of the world profoundly influence adult behavior, are delayed in primates and humans relative to mice[9–11].

However, a critical but unresolved question is whether the development of neuronal connections, the putative substrates of cognition and experience-dependent behavioral plasticity, is neotenous. Although synaptic development has been well studied[12,13], it has been studied with a myriad of approaches, ranging from electrophysiology[14], immunohistochemistry[15,16], gene transcription levels[17], transynaptic viral tracing[18,19], light level reconstructions of neurons (i.e., dendritic spines, dendrites, axons)[20,21], and single section EM[22–25]. However, each methodology has different false positive and negative rates, sometimes unknown, and it remains difficult to quantitatively relate developmental trajectories across species. This lack of a "ground truth" for describing connectivity has likely contributed to differing results over the extent and nature of synaptic development in model systems, such as the primary visual cortex of primates[26–28].

We asked whether an unbiased and complete reckoning of synaptic development across two species with disparate lifespans and disparate cognitive abilities would show differences in the rate and timing of post-natal synaptic development. In other words, would synapses in *Rhesus macaque* (primate) brains take longer to develop than in *Mus musculus* (mice)? A better understanding of synaptic neoteny would provide invaluable data on evolutionary innovations in brains across species, and how much of cortical circuitry is innate versus developed. Finally, understanding how synapses develop in primates and mice would point to potential mechanisms of how human brains develop.

We used large volume serial electron microscopy ("connectomics") to reconstruct excitatory and inhibitory connections onto excitatory neurons from multiple cortical regions (primary somatosensory (S1) and primary visual (V1) in mice and V1 in primates), from multiple cortical layers (Layers 2/3 and 4), across multiple time points post birth (p7 to p523 in mice, p7 to p3000 in primates), across multiple animals (*n* = 11 mice and *n* = 3 primates),

[1]Department of Neurobiology, The University of Chicago, Chicago, USA. [2]Argonne National Laboratory, Biosciences Division, Lemont, USA. ✉e-mail: gwildenberg@uchicago.edu; bobbykasthuri@uchicago.edu

and using a combination of publicly available and newly collected data sets. We chose EM as it remains the 'gold standard' for detailing neuronal connections and recent connectomic reports have revealed species differences[29,30] and developmental differences in connectivity[31]. Finally, we developed an algorithmic pipeline on national lab supercomputers for automated segmentation and 'saturated' tracing of neurons to verify these results[32,33].

Here we show that synapse development in mice and primates is isochronic, contrary to what models of neoteny would predict for long-lived, highly cognitive species like primates. We further show evidence of differences in excitatory and inhibitory synapse development between these species marking a potentially important distinction in how the brains of these two species change across their lifespans. Finally, we suggest, using prior data on human and chimpanzee, that neotenous synapse development may have evolved in hominids.

## Results

### Net excitatory synapse formation in the developing V1 and S1 of mice and V1 of primates

We first analyzed synaptic density on excitatory neurons in V1, L2/3 and L4 of mice and primates. We defined synapses as locations where a pre-synaptic axonal bouton with a vesicle cloud (~30–40 vesicles) was physically proximate to an identified dendrite or soma and contained a dark contrasted postsynaptic density (PSD). We classified synapses as excitatory or inhibitory on the basis of whether they synapse onto spines versus shafts and somata, respectively[29,34–36]. We started our first reconstructions at ~p6 in the mouse, a time when in other systems (e.g., the developing neuromuscular junction (NMJ) and autonomic ganglia of the peripheral nervous system (PNS)), individual post-synaptic neurons receive numerous pre-synaptic inputs[37–39]. Moreover, the first post-natal week in mice is prior to eye opening (~p14)[40,41], a period that has been reported to contain abundant, albeit potentially weak, synaptic connections among cortical neurons[42–44].

Our first surprise was the near absence of synapses in the early postnatal life of the mouse. At p6, mouse excitatory neurons in L2/3 and L4 were nearly absent of synaptic innervation (e.g., across two reconstructed neurons totaling ~125 μm of dendrite and complete soma, we found *zero* excitatory spine synapses, 89 dendritic filopodia, 1 somatic synapse, and 8 dendritic shaft synapses) (Fig. 1a, top panel, and bottom inset). We next evaluated spine and shaft synapse density across randomly sampled dendrites of varying diameter and orientation (e.g., dendrites likely from different parts of a neuron's dendritic tree) to ensure our analyses are not biased, for example, toward proximal dendrites. Consistent with our initial observations, we found mouse excitatory neurons at this age were sparsely innervated regardless of dendrite diameter or orientation (mean ± sem excitatory synapse density/μm of dendrite, mouse p6 L2/3 = 0.01 ± 0.007, L4 = 0.02 ± 0.008, $n = 20$, 10 μm dendrite fragments/dataset) (Fig. 1b, red lines, zoomed right inset). In fact, of the 40 randomly sampled 10 μm dendrite fragments in p6 mouse excitatory neurons in L2/3 and L4, only 6, contained spine synapses, and in those cases, mostly one. We confirmed the sparsity of synapses by measuring the synapse density of single, 2D EM sections from 5 additional p6 mice and found a mean ± sem synapse density of 0.465 ± 0.03 synapses/μm² in L2/3 and 0.428 ± 0.03 synapses/μm² in L4. This amounted to ~8.5–10x fewer synapses than p105 mice (mean ± sem synapse density: L2/3 = 4.84 ± 0.34 synapses/μm², L4 = 3.6 ± 0.32 synapses/μm²).

Primate neurons at p7 also showed sparse synaptic innervation relative to adults[29]. The primate p7 neuron reconstructed in Fig. 1a had 215 total synapses (i.e., spine, shaft, and soma) and 92 filopodia over ~100 μm of dendritic tree and complete soma reconstructed. While still lower than adult primate neurons, p7 primate excitatory neurons were qualitatively more spinous than similarly aged mouse neurons, with a mixture of filopodia and fully formed spine synapses (92 filopodia, 82 spine synapses). Excitatory neurons in p7 primates

had ~40x more spine synapses than p6 mice (e.g., mean ± sem spine synapses/μm, mouse p6 L2/3 | L4 = 0.01 ± 0.007 | 0.02 ± 0.008 vs primate p7 L2/3 | L4 = 0.31 ± 0.05 | 0.65 ± 0.08; L2/3 $p$ = 2.86e-7, L4 $p$ = 3.8e-7). The few spine synapses that could be identified in mouse p6 neurons appeared fully formed (i.e., a clear PSD and numerous synaptic vesicles, similar to primate p7 neurons) (Supplementary Fig. 1a). Finally, we found a correlation between dendrite diameter and synaptic density, but the effect size was small and not significantly different (Supplementary Fig. 2).

The differences in early innervation of primate and mouse cortical neurons prompted us to ask how synaptic numbers changed over development. Thus, we extended our analyses across multiple time points in L2/3 and L4 across both species, including time points covering the 'critical period' in the mouse cortex (i.e., p14 to p36)[9]. We found instead a seemingly monotonic increase in synapse density in both primates and mice over the first months of postnatal life (Fig. 1b).

As expected, primate neurons peaked in spine synapse density at age p75[22] relative to both p7 and p3000 ages. Notably, mouse excitatory neurons showed only a modest increase in the number of spine synapses from p6 to p14 (mean ± sem spine synapses/μm V1 L2/3 | L4: p6 = 0.007 ± 0.0065 | 0.02 ± 0.008, p14 = 0.87 ± 0.08 | 0.77 ± 0.09, $p$ = 1.1e-8 | 5e-8) and such densities continued to increase to age p105, peaking around the same absolute time as the primate. Like mouse V1, we next found mouse L2/3 and L4 neurons in the primary somatosensory cortex (S1) showed a similar sparsity at early postnatal life and a steady, seemingly monotonic, increase in the total number of excitatory synapses across postnatal life (Fig. 1b, black lines), suggesting that net synaptic growth over postnatal development was not unique to mouse V1. Overall, spine synapse density was broadly lower in baby and juvenile mice (e.g., mean ± sem spine synapses/μm for V1, L2/3: p6 = 0.01 ± 0.007, p14 = 0.87 ± 0.08, p36 = 1.63 ± 0.11 versus p105 = 2.1 ± 0.21), whereas juvenile primates had a higher spine synapse density compared to the adult (e.g., mean ± sem spine synapses/μm for V1, L2/3: p75 = 2.44 ± 0.27 vs. p3000 = 0.86 ± 0.09).

Finally, the similarity in the rise of excitatory synaptic density across mice and primates in absolute days prompted us to ask whether mouse neurons would show evidence of net pruning, later life, during a time frame, in absolute days, when primates also showed synaptic pruning. Thus, we analyzed similar synaptic density measurements in older mice, p523, ~1.5 years of age. Indeed, we found that excitatory synapse density drops in older mice (Fig. 1b), as previously reported in other brain regions and layers[45]. We found that both primate and mouse neurons added and pruned excitatory synapses at similar times in absolute days—one 'clock' for synaptic development across species with disparate lifespans. In contrast, we observed a subtle, though at times not statistically significant, increase in the number of putative inhibitory shaft inputs onto the same randomly sampled dendrite shafts across all V1 datasets (Fig. 1c, and Supplementary Fig. 1b) consistent with previous reports on postnatal changes of inhibitory shaft synapses in mouse S1[31]. Additionally, because ~20% of excitatory shaft synapses are made by excitatory axons[46,47], we checked whether the percentage of excitatory inputs onto excitatory dendritic shafts changed in V1, L2/3 across ages and species. We found the percentage of excitatory inputs onto shafts to be similar across ages, in both mice and primates (i.e., innervation patterns of shaft synapses are relatively unchanged over development) (excitatory synapses onto excitatory shafts: mouse p14 = 23% and p105 = 26%, primate p75 = 26% and p3000 = 26%, $n$ = 30 axons/dataset). Overall, we conclude:

- Little evidence of supernumerary increases in excitatory spine synapses followed by pruning during neonatal life in mouse.
- Rather, the rise and fall of excitatory spine synapses occur at approximately the same absolute time after birth in both species.
- In both species, the frequency of shaft synapses appears to slowly rise over the life of the animal with little change in the composition of excitatory and inhibitory innervation of shafts.

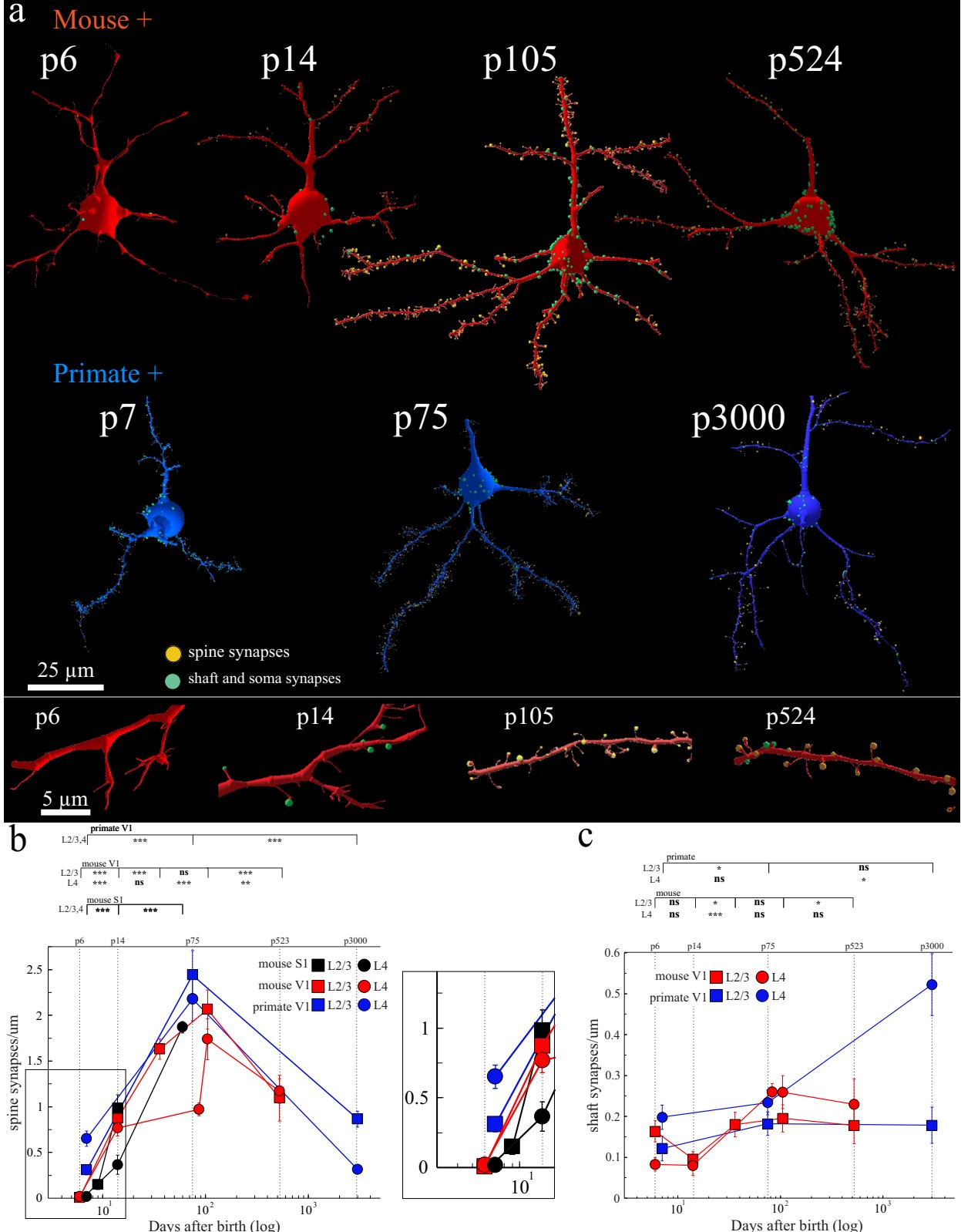

## Excitatory axon development

We next asked how excitatory axons changed during the periods of increasing and decreasing spine synaptic density we observed. We focused these analyses on L2/3 since developmental changes in spine synapse densities were equivalent between L2/3 and L4 in both species (i.e., Fig. 1). We considered two, but not mutually exclusive, mechanisms: (1) existing axons could increase their synapse frequency and/or (2) axons could locally branch more (Fig. 2a). We traced identified excitatory axons and annotated every axonal synapse and branch point made in our volumes at ages in mice and primates with large increases in net synapses (p14 to p105 in mouse and p7 to p75 in primates). Synapse density along axons increased

**Fig. 1 | Isochronic development of excitatory synapses in primate and mouse cortex. a** Shown are representative reconstructions of V1 mouse (top, red) and primate (bottom, blue) excitatory (+) neurons at the noted postnatal (p) days. Excitatory spine synapses = orange dots, inhibitory shaft and somatic synapses = green dots. *below:* zoom-ins of mouse excitatory dendrites from ages p7, p14, p104, and p524, left to right. **b** Scatter plot of x: postnatal days after birth (log) and y: average spine synapse density in synapses/μm. Top line ~ postnatal (p) days. Squares = L2/3, circles = L4, black = mouse S1, red = mouse V1, blue = primate V1. *Right:* close-up of earliest data points: p6-p14 mouse and p7 primate. **c** Scatter plot of x: postnatal days after birth (log) and y: average dendrite shaft synapses/μm. Top line ~postnatal (p) days. Two-tailed Mann-Whitney U test, ns = $P > 0.05$, * = $P < 0.05$, ** = $P < 0.01$, *** = $P < 0.001$ shown for pairwise comparisons between adjacent ages in each plot. See Supplementary Table 1 for numerical summary and supplementary files for all pairwise *p*-values. Lines that connect datapoints in scatter plots (**b**, **c**) are for visualization purposes only and do not represent fitted curves. Scale bar = 25 μm (**a**, top), 5 μm (**a**, bottom). Mouse p36 L2/3 and p87 L4 results derived from reanalyzing publicly available dataset: https://www.microns-explorer.org/. Mouse S1 results derived from reanalyzing publicly available datasets: p9 and p14[31] and p60[104]. *n* = 15–20, 10 μm dendrite fragments/dataset and 1 animal/dataset. Source data are provided as a Source Data file.

over this period in both species (Fig. 2b) (mean ± sem synapses/μm for mouse: p14 = 0.71 ± 0.09, p105 = 1.2 ± 0.07, *p* = 8.5e-9 and primate: p7 = 0.93 ± 0.1, p75 = 1.0 ± 0.09, *p* = 0.38, *n* = 50 axons/dataset).

However, the increase in synaptic density on axons could not account completely for the 2.4x and 7.9x increased number of synapses for mouse and primate, respectively. Thus, we asked whether local axonal branching could also contribute to increased synapse numbers. Indeed, we found that axons branched more over this period and in both species, there was a small population of axons increasing branch numbers dramatically (Fig. 2c) (mean ± sem branches/μm for mouse: p14 = 0.12 ± 0.02, p105 = 0.22 ± 0.02, *p* = 7.3e-3 and primate: p7 = 0.16 ± 0.03, p75 = 0.38 ± 0.04, *p* = 1.9e5, *n* = 50 axons/dataset). Thus, both increased synapse density along axons and axonal branching seem to contribute to increases in excitatory synapse density on neurons (i.e., Fig. 1). In both species, during periods of synaptic pruning, we find multiple examples of axons ending in a large bulb-like structure reminiscent of axon retraction bulbs in the mouse developing neuromuscular junction (NMJ)[48,49]: a sudden swelling at the end of an axon containing dense tortuously packed mitochondria (Fig. 2d, e). Furthermore, we found numerous examples where a branch of an axon would end in such a bulb while other branches of the same axon made clear synapses on postsynaptic targets (Fig. 2f), suggesting that such retraction is branch-specific. Bulbs were most notable at mouse p523 and primate p75, where we found sharp decreases in synaptic density. In both species, periods of synaptic pruning were associated with a reduction in synapse density along axons and decreased branching (Fig. 2b, c) (i.e., synaptic pruning likely occurs for both *en passant* and *terminal* synapses) (mean ± sem synapses/μm for mouse: p105 = 1.2 ± 0.07, p523 = 0.86 ± 0.09, *p* = 1.6e5 and primate p75 = 1.0 ± 0.09, p3000 = 0.46 ± 0.05, *p* = 2.7e-7, mean ± sem branches/μm for mouse: p105 = 0.22 ± 0.02, p523 = 0.15 ± 0.02, *p* = 7.0e-2 and primate p75 = 0.38 ± 0.04, p3000 = 0.22 ± 0.03, *p* = 4.7e-3, *n* = 50 axons/dataset). Lastly, given the sparsity of synapses in mouse p6 brains, we asked whether axons at this age made any synapses. We found that p6 axons made synapses at a similar frequency to other ages but formed very few branches (mean ± sem synapses/μm = 1.09 ± 0.13 and branches/μm = 0.09 ± 0.03, *n* = 50 axons). Moreover, boutons did contain vesicles (mean ± sem vesicles/bouton = 27.4 ± 2.6) though on average the number of vesicles was about 10-fold less than what has been reported in adult mice[36,50,51] (Supplementary Fig. 3). We did not classify these axons as excitatory or inhibitory as there were too few spine synapses at this age to use our axon classification system (See "Methods").

## The development of somatic inhibition across species

We next examined the development of somatic innervation of excitatory neurons in these two species. We restricted our analyses to somatic innervation of excitatory neurons which is almost exclusively made by parvalbumin (PV) interneurons in both species[52–54]. As we captured complete soma in our EM datasets, we assessed the total somatic synaptic contribution by PV innervation. First, similar to excitatory spine synapses, we find a complete sparsity of somatic innervation shortly after birth (Fig. 3a and see Supplementary Fig. 1c).

For example, the p7 primate excitatory soma in Fig. 3a had 24 synapses, and remarkably, the p6 mouse soma only had one, far less than what we and others have reported on adult somata from the same layers and regions[29,55,56]. We are aware that post-synaptic targets of inhibitory synapses are harder to identify in EM datasets as they lack clear post-synaptic densities. Thus, we use rigorous and specific criteria for their annotation (see "Methods", Supplementary Fig. 1b, c, and Supplementary Fig. 4 for examples). Over development, we find primarily addition of somatic synapses from p6 to p105 in mice and from p7 to p75 in macaques (Fig. 3b) (mean ± sem # synapses/soma; mouse L2/3 | L4, p6 = 1.2 ± 0.3 | 2.2 ± 1.0, p14 = 25.8 ± 3.5 | 11.3 ± 1.4, p36 = 66 ± 3.8 | 38 ± 0.82, p105 = 72.2 ± 5.1 | 42.9 ± 4.6; primate L2/3 | L4, p7 = 15.8 ± 2.1 | 23.5 ± 2.7, p75 = 50.3 ± 4.1 | 32.7 ± 4.9, *n* ≥ 6 soma/dataset). Again, like excitatory connections, the similarity in synaptic addition on somata across mice and primates prompted us to check somatic synaptic density at older ages. At later ages, we saw our first difference in synaptic development across species. The number of inhibitory synapses on primate excitatory soma reduced in both L2/3 and L4 of V1 from ages p75 to p3000 (Fig. 3a, b, blue soma reconstructions and data points) (mean ± sem #synapses/soma for primate L2/3 | L4, p75 = 50.3 ± 4.1 | 32.7 ± 4.9, p3000 = 16.1 ± 2.1 | 14.2 ± 1.4, *p* = 2.366e-3(L2/3), *p* = 4.77e-3 (L4)). Unlike primate, mouse L2/3 and L4 neurons gradually increase, or perhaps plateau, in their number of somatic synapses by the latest age we sampled (i.e., p523) (Fig. 3a, b, red somata reconstructions and data points) (mean ± sem #synapses/soma for mouse L2/3 | L4, p105 = 72.2 ± 5.1 | 42.9 ± 4.6, p523 = 79.2 ± 7.1 | 53.8 ± 7.5, *p* = 0.6(L2/3), 0.1(L4)). We next reconstructed the inhibitory axons making these somatic synapses in L2/3. For both species, we found the increased soma synapse formation was the result of more axons innervating soma rather than the number of somatic synapses per axon increasing (Fig. 3c) (mean ± sem #axons/soma: mouse L2/3 p14 = 17.25 ± 2.5, p105 = 33 ± 2.5, *p* = 0.016, primate L2/3 p7 = 13.3 ± 2.7, p75 = 22.4 ± 1.6, *p* = 0.048, *n* = all soma-innervating axons across ≥6 soma/dataset). Likewise the number of axons innervating soma declined in primate from p75 to p3000 but remained the same, if not slightly increased, in mice from p105 to p523 (mean ± sem #axons/soma: mouse L2/3 p105 = 33 ± 2.5, p523 = 39.3 ± 1.0, *p* = 0.11, primate L2/3 p75 = 22.4 ± 1.6, p3000 = 12.5 ± 1.5, *p* = 0.013, *n* = all soma-innervating axons across ≥ 6 soma/dataset). Thus, synaptic pruning on primate excitatory neuronal somatas, and not in mice, is axonal pruning (i.e., not simply pruning of synapses while maintaining the numbers of innervating axons). Indeed, for primate excitatory neurons we see a dramatic ~50% increase and reduction in the number of inputs per soma from ages p7 to p75 and p75 to p3000, respectively.

Our discovery of inhibitory axonal pruning on excitatory somata of primate neurons (summarized in Fig. 3d) prompted us to investigate the consequences of this process. We found that after pruning, the size of the remaining boutons making somatic synapses on primate excitatory neurons grew significantly (Fig. 3e and Supplementary Fig. 4) (mean ± sem bouton volume, primate L2/3 p7 = 0.08 ± 0.01, p75 = 0.16 ± 0.02, p3000 = 0.27 ± 0.03, *n* ≥ 40 boutons/dataset, p7 vs p75, *p* = 3.7e-5, p75 vs p3000, *p* = 5.1e-4). Indeed, changes in bouton volume between primate p7 and p3000 were qualitatively apparent in our 3D reconstructions when the surface

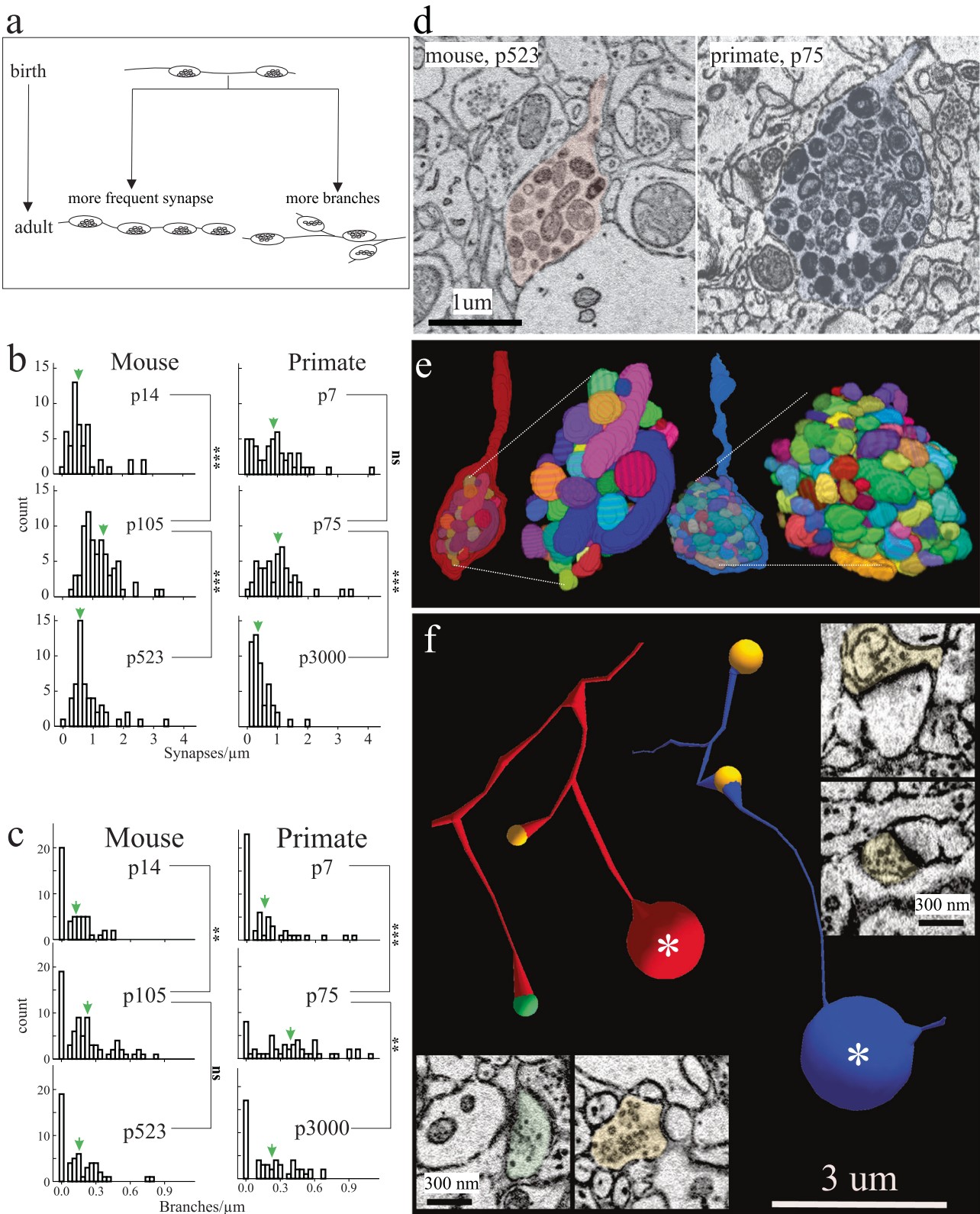

**Fig. 2 | Excitatory axon development in mouse and primate. a** Cartoon depicting hypothetical models of excitatory axon development: *left*, axons increase their synapse frequency and/or *right*, axons make more branches. **b**, **c** Histograms of the number of synapses/μm and branches/μm, respectively, of excitatory axons at different ages in mouse (left) and primate (right) in V1 L2/3. Green arrows indicate the ~mean. **d** Single 2D EM images and **e** 3D reconstructions of a representative terminal axon retraction bulbs in mouse V1, p523 (left) and primate V1, p75 (right). 3D reconstructions show individually colored mitochondria contained within the retraction bulb. **f** Skeleton reconstructions of mouse (red) and primate (blue) axons containing terminal retraction bulbs (asterisk) (from **d**) and spine (orange circle) or shaft (green circle) synapses. *Insets*: 2D EM images of spine and shaft synapses made by the primate (right) and mouse (bottom) axon containing a retraction bulb. Two-tailed Mann-Whitney U test, ns = $P > 0.05$, * = $P < 0.05$, ** = $P < 0.01$, *** = $P < 0.001$ shown for pairwise comparisons between adjacent ages in each plot. See Supplementary Table 2 for numerical summary and supplementary files for all pairwise *p*-values. Scale bar = 1 μm (**d**), 3 μm (**f**) and 300 nm (**f**, right and bottom insets). Source data are provided as a Source Data file.

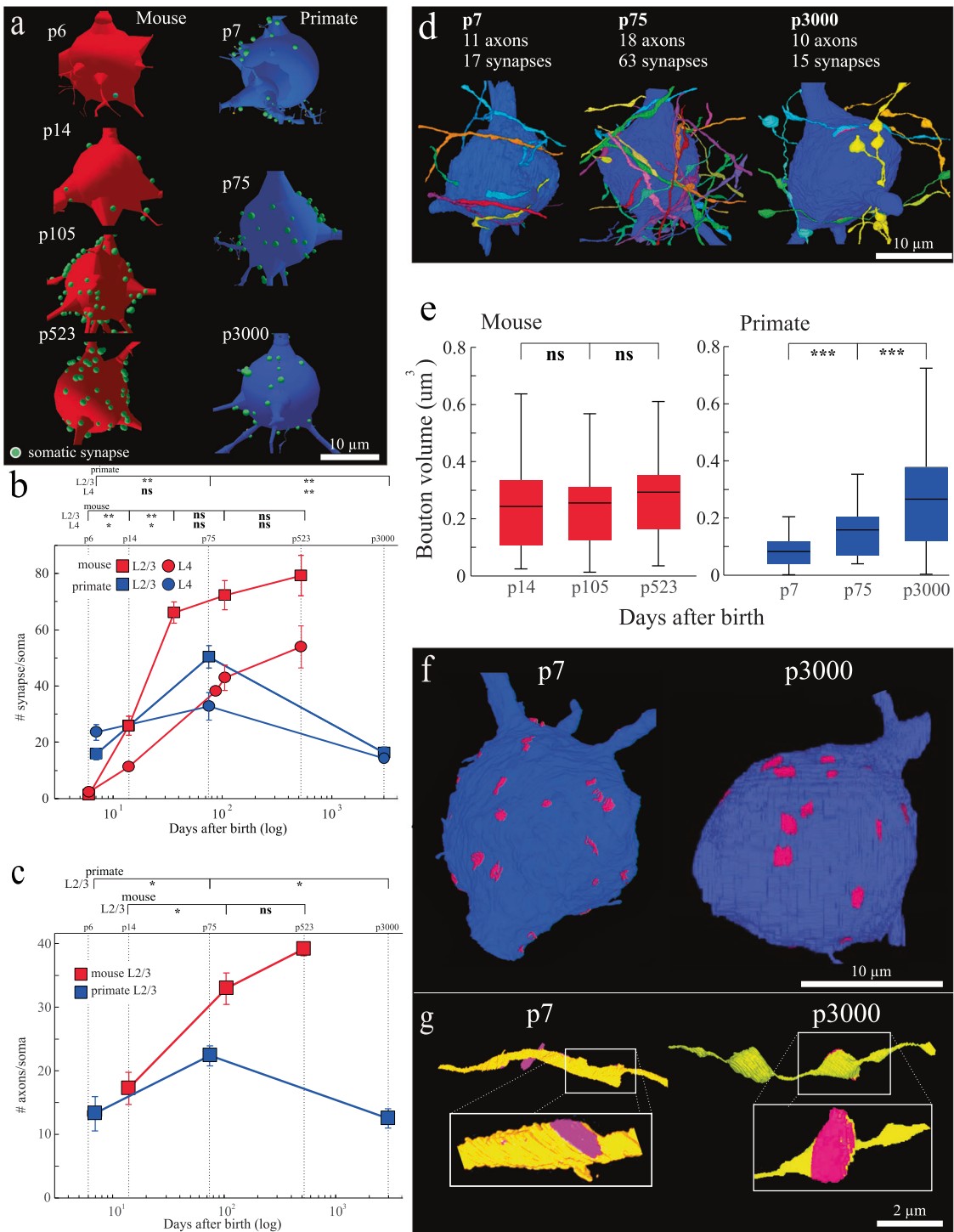

**Fig. 3 | Soma innervating inhibitory axons develop differently in mouse and primate. a** Representative reconstructions of mouse (left, red) and primate (right, blue) V1, L2/3 excitatory somata at the noted postnatal ages. Green dots mark the positions of all soma synapses on each neuron. **b** Scatter plot of x: postnatal age (log) and y: the total number of somatic synapses/soma. Blue lines = primate, red lines = mouse. Squares = L2/3, circles = L4. Top line ≈ postnatal (p) days. **c** Scatter plot of x: postnatal days (log) versus y: the total number of innervating axons/soma for primate (blue) and mouse (red) V1, L2/3 excitatory neurons. Top line ≈ postnatal (p) days. **d** Representative 3D reconstructions of excitatory soma and soma-innervating axons from primate V1 L2/3. Each reconstruction lists the postnatal age, total number of innervating axons, and total synapses found on the depicted soma. **e** Box plot of mouse (left, red) and primate (right, blue) V1, L2/3 soma synapse

bouton volume. Black lines = mean, boxes show the interquartile range and lines/whiskers define the min/max value. **f** Representative 3D reconstruction of V1, L2/3 excitatory soma from p7 and p3000 primate marking PSD (pink) of each somatic synapse. **g** Close-up of 3D reconstruction of one soma-innervating axon from p7 and p3000 depicting the qualitative difference in PSD size. Two-tailed Mann-Whitney U test, ns = $P > 0.05$, * = $P < 0.05$, ** = $P < 0.01$, *** = $P < 0.001$ shown for pairwise comparisons between adjacent ages in each plot. See Supplementary Table 2 for numerical summary and supplementary files for all pairwise $p$-values. Lines that connect datapoints in scatter plots (**b**, **c**) are for visualization purposes only and do not represent fitted curves. Scale bar = 1 μm (**e**), 3 μm (**g**) and 300 nm (**g** insets). Source data are provided as a Source Data file.

area of the PSD was overlayed onto the post-synaptic soma (Fig. 3f, pink areas = PSD 3D reconstructions, and 3g, representative axon (yellow) with one depicted PSD in pink). The size of mice somatic synapses, which showed little sign of pruning, remained relatively unchanged over the period we examined (Fig. 3e). These results suggest, like in other systems with synaptic pruning[37,49], 'surviving synapses' become strengthened by becoming larger[57]. We also note that the total synaptic occupancy of the soma at any age was qualitatively less than ~10% of the total soma surface (e.g., see Fig. 3d, f). Thus, it is unlikely that pruning is driven by competition for limited physical space. These results collectively demonstrate evidence of inhibitory synaptic pruning in the primate but not the mouse, indicating that the synaptic development of somatic inhibitory inputs may mark an important evolutionary distinction between these two species.

### Saturated reconstruction and comparison between mouse neonate (p14) and adult (p105) V1

We next performed algorithmic assisted 'saturated' reconstruction of the developing mouse brain to:

- More exhaustively analyze synaptic development in mouse cortex, given the surprise of finding little evidence of net synaptic pruning on postnatal mouse neurons.
- More quantitatively measure changes in synaptic size, map changes in non-synaptic morphologies like filopodia, and the development of sub-cellular organelles.
- Provide a resource of large volumes and annotations to the community (see "Methods" for data sharing plan).

Thus, we scaled and parallelized on Argonne National Laboratory supercomputers, a custom algorithmic pipeline for creating 3D EM volumes, tracing neuronal processes and identifying their connections, and incorporating human error checking (Supplementary Fig. 5). We used this algorithmic pipeline to analyze 91 neurons, 195 of their dendrites, and 551,652 synapses (of which 20,318 were manually proofread) on L4 mouse neurons at p14 and p105. Specifically, we utilized a combination of a flood filling network (FFN)[58] for saturated segmentation of neurons and UNet combined with 3d connected components (i.e., watershed)[59–63] to segment synapses and mitochondria for quantifying developmental changes in synapse size, filopodia, and mitochondrial number, morphology and size. We proofread 88 dendrites from the p14 dataset originating from 41 cell bodies and 107 dendrites from the p105 dataset from 50 cell bodies, overall proofreading 6,809 synapses at p14, 13,509 synapses at p105 (Fig. 4a–c). Overall, the cable length of fully annotated dendrites in p14 data reached 15.4 mm and in p105 reached 19.5 mm.

### Further evidence of primarily synapse formation in developing mouse visual cortex

We observed a mean ± sem synapse density of 0.43 ± 0.11 synapses/μm across 88 dendrites at p14 and 0.69 ± 0.16 synapses/μm across 107 dendrites at p105, amounting to a 60.4% increase ($p = 6.17e26$) (Fig. 4d). We believe the differences from the automated segmentation data and our hand tracings (i.e., Fig. 1b) are likely due to an over-representation of large proximal dendrites attached to soma in our automated analyses (see Supplementary Fig. 2). We also manually annotated filopodia along these dendrites and found a significantly higher density in p14 data, while in p105, filopodia are extremely rare (mean ± sem: p14 = 0.075 ± 0.038, p105 = 0.005 ± 0.008, $p = 4.43e-32$) (Fig. 4e). We used the distinct morphological criteria of filopodia as long protrusions from dendrite that do not form a postsynaptic structure[64,65]. Finally, we also found a similar rise in the number of somatic synapses from p14 to p105 (mean ± sem: p14 = 18.31 ± 12.15, p105 = 42.85 ± 28.80, $n = 39$ and 41 soma, respectively, $p = 2.65e-8$) (Fig. 4f). Overall, these results from automatic segmentation are consistent with and provide further validation of our manual annotation.

### Increase in synapse size

A major advantage of connectomics is its ability to resolve ultra-structural neural morphology. With a combination of automatic mask prediction and manual correction, we were able to quantitatively measure each synapse for its vesicle size and PSD size (Supplementary Fig. 6). We compared the distribution of these metrics at p14 and p105 and observed a significant increase in both synaptic junction size (66.5%) and in vesicle cloud size (14.8%) in p105 mice (both $p < 1e-5$) (Supplementary Fig. 7), suggesting an overall maturation of synapses. Notably, synapse size distributions are primarily log-normal, similar to synapse size distributions in the adult mouse brain[66,67], suggesting that such distributions are not the result of developmental synapse re-arrangements. Thus, we conclude that the primary development of synapses in mouse primary visual cortex is the addition of new synapses and their growth with the concurrent removal of large numbers of filopodia. Finally, we find that the average distance of a spine synapse from 'parent' dendrites is similar at both p14 and p105 (Supplementary Fig. 8) (mean ± sem distance to dendritic branch (nm); p14 = 1107 ± 9.7, p105 = 1081 ± 7.5, $p = 0.80$) despite a 60.4% increase in the number of synapses, suggesting intrinsic limitations to the distances spines can extend, independent of the number of synapses formed and unaffected by the process of development.

### Mitochondria size development and correlation with synapse density

Our segmentation pipeline allows us to reconstruct sub-cellular organelles in addition to synapses, permitting correlations between the development of organelles and the development of synapses. As an example, we investigated correlations between mitochondria and synapse development. We chose mitochondria as they have been implicated in numerous synaptic functions including sustaining long-term plasticity[68–70]. We reconstructed 398,278 instances of mitochondria in p14 and 533,019 in p105. We analyzed mitochondria primarily in the dendrites of excitatory neurons and found a 78.4% increase in mitochondria size and 66.7% increase in mitochondria density (i.e., total mitochondria volume/length of dendrite analyzed) in p105 relative to p14 (Supplementary Figs. 9, 10) (mean ± sem mitochondria size ($μm^3$); p14 = 0.04263 ± 0.0, $n = 39,8278$, p105 = 0.0687 ± 1.9e-4, $n = 533,019$, $p ≈ 0$; mean ± sem mitochodria density ($nm^2$); p14 = 19,236.2 ± 571.1, $n = 88$, p105 = 32,072.9 ± 2,581.3, $n = 107$, $p ≈ 1e-20$). Finally, previous connectomic analyses have demonstrated a positive correlation between synapse density and mitochondria coverage in basal dendrites of neurons in the mouse V1, L2/3[71]. We found, indeed, that this correlation emerges with developmental age—at age p105, there is a stronger correlation between mitochondria coverage and synapse density in basal dendrites relative to p14, but little correlation in apical dendrites in both ages (Supplementary Fig. 11) (Pearson correlation coefficient, apical and basal combined: p14 r = 0.12, p = 0.25, p105 r = 0.5, p = 5.8e-8; apical: p14 r = −0.01, p = 0.97, p105 r = 0.18, p = 0.44; basal: p14 r = 0.2, p = 0.1, p105 r = 0.54, p = 6.4e-8).

## Discussion

Despite their obvious difference in life span, cognition, etc., we found little evidence that synaptic development, excitatory or inhibitory, was delayed, or neotenous, in primates relative to mice. (Supplementary Fig. 12). Instead, excitatory synapse development was isochronic (Fig. 1b)—synapses were added and eliminated at similar rates and at similar absolute times after birth in both species. A lack of pruning in developing mouse inhibitory connections onto the soma of excitatory neurons was the one difference across species. We reached several conclusions from these results.

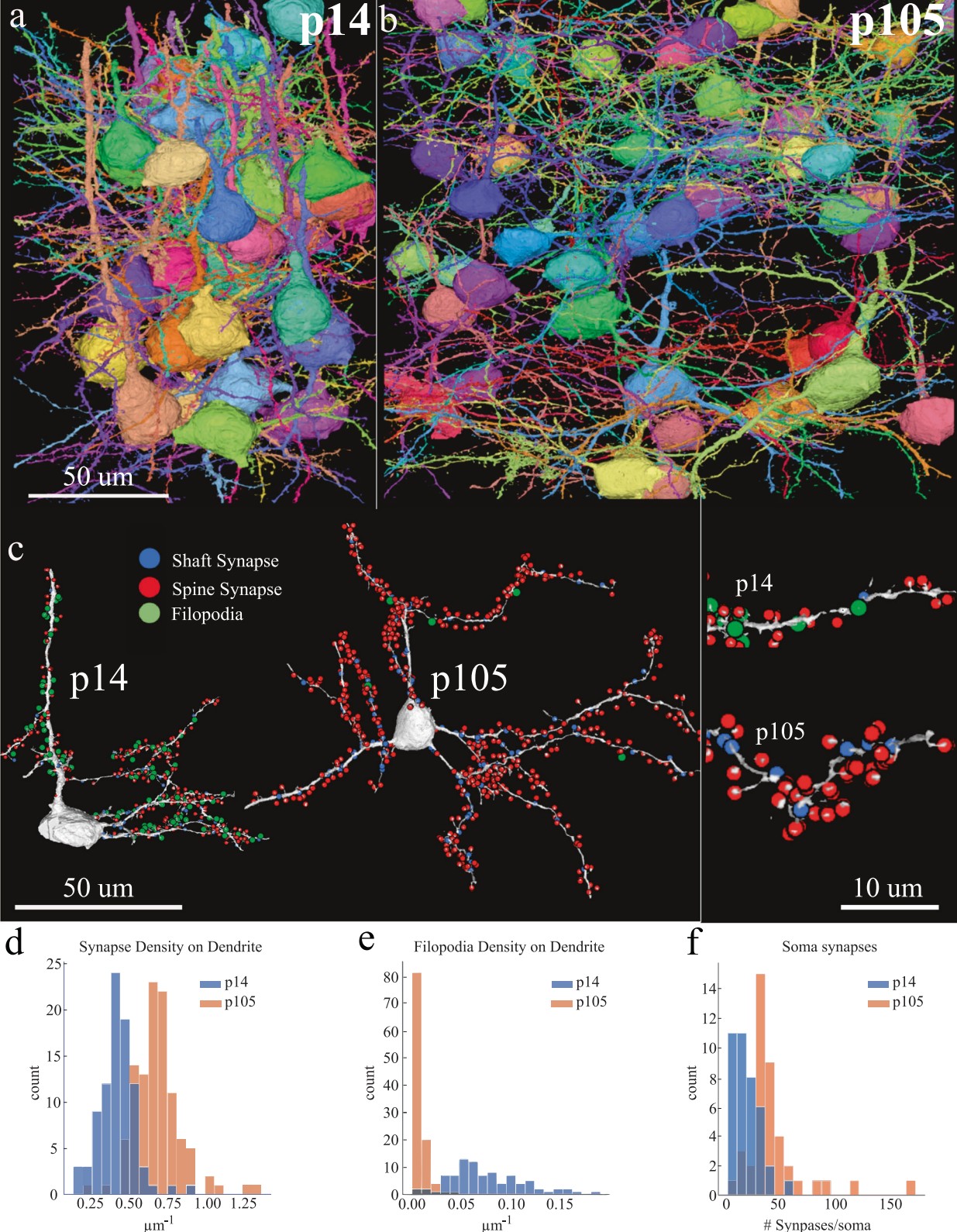

**Fig. 4 | Automatic segmentation of mouse V1 p14 and p105 excitatory neurons.**
**a**, **b** Representative reconstructions of L4 excitatory neurons from mouse V1, p14, p105. Each neuron is uniquely colored for visual purposes. **c** Representative reconstruction of a single mouse V1 p14 and p105 excitatory neuron with the position of inhibitory shaft synapses (blue dots), excitatory spine synapses (red dots), and filopodia (green dots) detected using automatic segmentation. *Right*, zoom in of left images showing a dendrite from p14 (top) and p105 (bottom).

**d**–**f** Histograms of mouse V1 p14 (blue) and p105 (orange) comparing distribution in the frequency of **d** excitatory spine synapses/μm, **e** filopodia/μm, and **f** total number of inhibitory synapses/soma. Two-tailed Mann-Whitney U test; $p = 6.17e\text{-}26$ (**d**), $4.43e\text{-}32$ (**e**), and $2.65e\text{-}8$ (**f**). Scale bar = 50 μm (**a**–**c**, left), 10 μm (**c**, right). mean ± sem can be found in the Source data file. Total automatically segmented and manually proofread synapse counts for (**d**–**f**): mouse p14 = 6809, p105 = 13509. Source data are provided as a Source Data file.

First, this is not the first demonstration of isochronic synapse development, as synapses across different primate cortical areas (e.g., pre-frontal, motor, primary visual cortices, etc.) also seem to add and prune at similar rates and at similar times[72]. Thus, our data extends these observations, suggesting a universal clock for synapse development across species and cortical areas independent of whether the world is experienced as a mouse or a primate. Furthermore, we find that not only are the mechanisms of synaptic development seemingly similar across cortices of different species but also across peripheral and central synapse development[37,49]. Like the NMJ, dendrites of primate excitatory neurons show sparse anatomical inputs at early ages, a dramatic increase in the number of axonal inputs followed by axonal/synaptic pruning. As in the NMJ, we find, during periods of net synaptic pruning, numerous examples of mitochondria-filled 'axonal retraction' bulbs (Fig. 2d–f), an anatomical signature of axonal pruning at NMJs. These similarities suggest that axonal retraction might be a common mechanism for synaptic rearrangement in the nervous system. Notably, our previous observations that such retraction bulbs are formed during exposure to drugs of abuse[73] suggest that this potential common mechanism of axon removal could be used during pathological synaptic pruning. Finally, numerous studies in peripheral systems have suggested that an outcome of synaptic pruning is that inputs that survive are strengthened. Indeed, inhibitory axonal pruning in primates was similar—somatic synapses that survive developmental pruning increased in size (Fig. 3e–g), which suggests that they were likely strengthened[57]. We conclude that the similarity of synaptic development across species and across peripheral and central synaptic development suggests a small number of regulators of the process, perhaps controlled a few conserved genes. Indeed, recent reports that neuronal activity during development is not necessary for the emergence of 'normal' neural behavior and neural circuits underscores our conclusion that the timing of synaptic development seems uncoupled from experience[74].

Second, the remarkable similarity in the time course of synaptic development of mice and primates contrasts sharply with the limited ultra-structural data available for humans. Albeit from sparse sampling of individual EM sections, such data reveals a profound delay in synapse formation and pruning in multiple cortical areas of developing human brains relative to the trajectories of macaque and mouse brains reported here (Supplementary Fig. 13). Thus, the potential neoteny of human synaptic development seems a more recent evolutionary adaptation than the divergence from a common ancestor with macaques (-34 million years ago[75]). Given the minimal change in developmental programs across mice and macaques, with more than 87 million years of evolutionary divergence[75], we argue that human neoteny seems to have emerged as a dramatic and potentially quantal evolutionary adaptation. One possibility, given the similarity of chimpanzee and human neuronal development[76], is that neoteny might have emerged among *Hominoidea* with a shared common ancestor with humans -6.5 mya[75]. Future investigations applying connectomics analyses to post-mortem developing human brains compared to the development of more primate species could help narrow down when the neoteny of synaptic development evolved. Finally, the seemingly quantal gap in the timing of synapse development between human and mouse/primate brains predicts the existence of a similarly large gap in the underlying control, potentially genetic, of that program. Indeed, our results suggest that comparative transcriptomic or other genetic analyses in the first three months of post-natal life among humans, macaques and mice brains could be an ideal moment to identify genetic programs for synapse formation present in mice and non-human primates (NHP) but not humans (e.g., the potential genetic mechanisms for human synaptic neoteny).

Third, our data, particularly reconstructions of the saturated automated segmentations from the developing mouse brain (Fig. 4), demonstrate little sign of net synaptic pruning, excitatory or inhibitory, in early mouse post-natal life, despite some reports to the contrary[25,77,78]. Our results favor a model of excitatory synapse formation in the cortex where filopodia mature into spine synapses[42,64,65,79–82] as opposed to other models including transformation of shaft synapses into spines, or spine formation followed by pruning (Supplementary Fig. 14). For example, we find that ~24% of shaft synapses are likely excitatory across developmental ages. Even if all shaft synapses converted to spine synapses, that would still not account for the increase in spine density we observed. Indeed, multiple single-section EM studies across a range of species (e.g., rat[83], rabbit[84], and cat[85]) all show little sign of net synaptic pruning in the first few months of post-natal life. The widespread prevalence of filopodia in p7-p14 neurons, which can be hard to differentiate from spines without unambiguous identification of a pre-synaptic axonal partner, could have potentially inflated counts of synaptic density in reports using lower resolution and sparse labeling optical microscopy. There are also numerous reports of changes, potentially heterochronic changes, in gene transcription[3,86,87], sometimes even of genes implicated in synapse formation and pruning[88]. However, since we are uncertain about the 'conversion' factor (i.e., how many additional RNA transcripts correlate to an additional synapse), we cannot relate our findings to those.

As we measured only net changes in synapses, we cannot speak to dynamic changes (i.e., synapses that are pruned or reformed), or if axons re-arrange such that the total number of distinct inputs to a post-synaptic neuron go down but the total number of inputs increases (e.g., synaptic take-over during input elimination at the developing NMJ[89]). Finally, we find little evidence of net pruning of inhibitory connections over early mouse postnatal life, extending previous results that used sparse sampling EM without identification of somatic vs. shaft inhibitory synapses[25]. Inhibitory synaptic pruning has been implicated as a potential mechanism for determining the end of functional critical periods in early mouse post-natal life[90–92]. The lack of evidence of anatomical pruning of inhibitory synapses suggests that functional properties of inhibitory synapses[92] (e.g., release probabilities[93–96], post-synaptic receptor composition[97–100], GABA transporter[101]) are more likely to underlie that change. Finally, it is possible that the differences in inhibitory development we observe across mice and macaques are a reflection of the fact that mice neurons have more excitatory connections in adult life relative to primates[29] (i.e., more excitatory synapses per neuron in mice require more inhibitory synapses to keep the neurons and circuits under control[102]).

## Methods

### Animal subject and tissue acquisition details

All experimental aspects conducted comply with The University of Chicago Institutional Animal Care and Use Committee (IACUC), Animal Resource Center and Office of Research Safety following approved animal protocol 72480. C57Bl/6 mice and *Rhesus macaque* were used in this study. The brain tissue processed in our lab (i.e., mouse V1 p6,p14,p105,p523 and primate V1 p7,p75,p3000) was prepared for EM as previously described[103]. Briefly, for mice, animals were deeply anesthetized using an intra-peritoneal (IP) injection of Pentobarbitol at 120 mg/kg until unresponsive to pinch in limbs and tail. Mice were then transcardially perfused first with buffer (0.1 M Sodium Cacodylate, pH 7.4) followed by fixative (0.1 M Sodium Cacodylate, pH 7.4, 2% paraformaldehyde, and 2.5% glutaraldehyde). The brain was then extracted and postfixed for 24 h in fixative at 4 °C. Brains were vibratome sliced in 300 μm thick coronal sections, a piece of brain spanning V1 was cut out with a scalpel, and stained with heavy metals, dehydrated and embedded in plastic for electron microscopy. Dissection of V1 and serial electron microscopy imaging was performed as previously described[29,104]. Primate p3000 tissue was acquired by our lab from a previous study[29] where the primate was first deeply anesthetized by intravenous (IV) administration of Pentobarbitol at 120 mg/kg until

unresponsive to pinch in limbs and tail. For primate tissue, the primate was initially sedated with Ketamine intramuscular (IM) at 3 mg/kg and dexmedtomidine IM at 75 mcg/kg. After sedation but prior to intubation, he was given buprenorphine SQ at 0.015 mg/kg. He was intubated and maintained on 1% isoflurane inhalant anesthesia at a surgical plane of anesthesia with a ventilator while the chest was incised and the sternum opened. The descending aorta and caudal vena cava were clamped. A 16 gauge needle was inserted into the left ventricle of the heart and the right atrium was cut with scissors. Heparinized saline was pressure-infused into the left ventricle until the blood was mostly cleared. Then 10% buffered formalin was connected to the needle and the brain was perfused. All procedures for primate perfusions were performed by the University of Chicago Animal Care veterinarian staff following approved protocols. Primate p7 and p75 tissue were kindly provided to us by Dr. Alvaro Duque of the MacBrain Resource Center (MBRC) of Yale School of Medicine. The MacBrain Resource Center is supported by NIH grant R01MH113257 (to Dr. Alvaro Duque).

### Data collected
The total volumes imaged and links to external datasets are listed in Supplementary Table 3.

### Manual segmentation
Data annotations were done using Knossos (https://knossos.app/) and performed by 3 individuals (G.W., H.L., B.K.). To ensure the accuracy of the data, 33% of the annotations from one person was verified by the other; 33% of the annotations were given to a naïve annotator to verify the accuracy. We found a >98% agreement between manual annotators. We found that 100% of spines could be reconstructed. Classes of neurons and their dendrites were identified by distinguishing anatomical properties[29]. We used the following metrics to identify and quantify each anatomical feature reported on: (1) Excitatory synapses were identified by the presence of a dendritic spine containing a post-synaptic density and vesicles on the pre-synaptic axon and whose membranes were in apposition to each other and no other touching neurite. (2) Soma and shaft inhibitory synapses: excitatory soma were identified as being a part of neurons with spinous dendrites. Neurons whose soma was fully within the imaged volume were used to count the total number of soma synapses. Perisomatic synapses were scored along the first 10 μm of dendrite that left the soma. Both somatic and shaft synapses were identified by finding a pre-synaptic, vesicle-filled bouton containing a flattened membrane in apposition to the post-synaptic membrane. By scanning in z around this area, a darkening of the area between the touching membranes was identified as a putative PSD. Additionally, the pre-synaptic bouton also did not show any membrane apposition to other neurites it was touching. (3) Spine and shaft synapse frequency: a dendrite was chosen at random and traced for 10 μm to first determine if it was an excitatory dendrite. The number of spine or shaft synapses contained within the ≈10 μm window were counted manually in Knossos and the total number of synapses were divided by the actual segment length to calculate spine synapses/μm and shaft synapses/μm. The diameter of the dendrite was calculated by measuring across the diameter in all three orthogonal views and then averaged. (4) Bouton size: a node (i.e., sphere) was placed over a bouton in Knossos and sized to best fit the size of the bouton. The node radius was used to calculate the surface area using S.A = $\pi 4r^2$. (5) Excitatory axons were identified as those making synapses with dendritic spines. (6) Excitatory synapses onto excitatory shafts: 30 boutons making shaft synapses onto excitatory dendritic shafts were randomly selected. The axons of those boutons were traced through the volume until the axon made at least 3 additional synapses. If any one of those synapses were with a dendritic spine, the axon was classified as excitatory.

### Bulk synapse density
We analyzed synapse density in single section EM in 5 non-contiguous sections in 5 additional animals in mice at p6. For each 2D section, we analyzed 4 randomly chosen 20 um² fields of view (FOV) with the only criteria that the FOV spanned dense neuropil and did not include any soma or blood vessels. Synapses were counted if there was a clear PSD and pre-synaptic vesicle cloud with at least 5 vesicles.

### 3D rendering
All 3D manual segmentation rendering was done using VAST[105] or neuroglancer.

### Statistical methods
Mean and standard error of the mean (SEM) were calculated for every quantification. Statistical significance was calculated using the two-tailed Mann-Whitney U test[106] between aggregate mouse and aggregate primate datasets. Mann-Whitney U was implemented in r using the Wilcoxon test. See Supplementary Tables 1 and 2 and figure legends for the exact value of (*n*) and what (*n*) represents. All pairwise *p*-values available in the Source Data File. Box plots indicate variability outside the upper and lower quartiles and the center black line indicates the mean.

### Reporting summary
Further information on research design is available in the Nature Portfolio Reporting Summary linked to this article.

## Data availability
All original EM data for V1 are freely available online at: https://bossdb. org/project/wildenberg2023 or DOI:10.60533/boss-2023-0s41. Mouse S1 datasets that we analyzed for changes in excitatory synapses are available at their original publication[31] which can be found at: https:// pubmed.ncbi.nlm.nih.gov/33273061/. Mouse p60, S1 excitatory synapse measurements were manually analyzed using the publicly available neuroglancer file from[104] which can be found here: https:// github.com/google/neuroglancer. Mouse p36 datasets are publicly available here for L2/3: https://www.microns-explorer.org/phase1 and here for, p87, L4: https://www.microns-explorer.org/cortical-mm3. Automatic segmentation of neurons, synapses and mitochondria of mouse V1, L4 p14 and p105 are available as a WebKnossos format for further public proofreading/error checking. For Figs. 1–3, source data can be found in the source data Excel file. All other source data (i.e., Fig. 4 and Supplementary Figs. 7–11) can be found here: https://bossdb. org/project/wildenberg2023. Source data are provided with this paper.

## Code availability
Homemade code used for EM image processing (i.e., 2D stitching, 3D alignments, brightness, and contrast normalization) can be freely accessed here: https://github.com/Hanyu-Li/klab_utils or https://doi. org/10.5072/zenodo.1246118[107] 3D alignment was performed using the publicly available Aligntk software available here: https://mmbios.pitt. edu/aligntk-home. Homemade code used for data analysis of Knossos traced skeletons (i.e., XML files) can be freely accessed here:https:// github.com/knorwood0/MNRVA or https://doi.org/10.5072/zenodo. 1246116[108]. An extensive description of all automatic segmentation details can be found here: https://knowledgeuchicago.edu/record/ 3579?ln=en and see Supplementary Fig. 5.

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

## Acknowledgements

Primate p7 and p75 tissue were kindly provided to us by Dr. Alvaro Duque of the MacBrain Resource Center (MBRC) of Yale School of Medicine. The MacBrain Resource Center is supported by NIH grant R01MH113257 (to Dr. Alvaro Duque). We thank The University of Chicago Animal Resources Center (RRID:SCR_021806), especially Jennifer McGrath, Alyssa Brown, Erika Becerra, and Marek Niekrasz, for their assistance with primate perfusions and general animal care. We would like to sincerely thank Vandana Sampathkumar, Dawn Paukner, and Anastasia Sorokina for helping with the manual proofreading of the automatic segmentation. We would also like to thank Dr. Kevin M Boergens, Professor Murray S. Sherman, Professor John Maunsell, and Professor Peter Littlewood for their thoughtful suggestions on manuscript preparation. This work was supported in part by funding from the NSF (award #2207383—G.W. and #2014862—N.K.).

## Author contributions

G.W. and N.K. conceived the idea, interpreted data, wrote the manuscript, and managed the editorial reviews. N.K. helped with confirming manual annotations by reanalyzing a significant fraction. H.L. collected datasets for mouse p14 and p105, L4 including mouse perfusions, staining tissue for EM, section collection and imaging. H.L. performed all automatic segmentation portions. A.S. and V.S. performed extensive proofreading of segmentation results from automatic segmentation generated by H.L. G.W. collected all other EM datasets for mouse and primate (i.e., tissue collection, staining, sectioning, imaging, alignment) and performed all the manual annotations for these data. G.W. pulled all public datasets and annotated manually.

## Competing interests

The authors declare no competing interests.
