## [Peer Review File · Nature Communications]

Isochronic Development of Cortical Synapses in Primates and MiceEditorial Note: This manuscript has been previously reviewed at another journal that is not operating a transparent peer review scheme. This document only contains reviewer comments and rebuttal letters for versions considered at *Nature Communications*. Mentions of the other journal have been redacted.

REVIEWERS' COMMENTS

Reviewer #2 (Remarks to the Author):

The authors have satisfactorily addressed all my questions, and I appreciate their responses. However, upon examination of their data availability section, I've discovered a error that I had previously overlooked.

The MICrONS dataset that contains layer 4 was collected from a P87 mouse, not a P36. While this error may not impact their overall conclusions. nevertheless, it does affect the figures and should be corrected

Reviewer #3 (Remarks to the Author):

I had reviewed this paper in its previous life at [REDACTED] and found it to be a remarkably interesting and well executed set of studies. The authors have responded the the very few concerns that arose then. As a result I am happy to recommend acceptance.

REVIEWERS' COMMENTS

Reviewer #2 (Remarks to the Author):

The authors have satisfactorily addressed all my questions, and I appreciate their responses. However, upon examination of their data availability section, I've discovered a error that I had previously overlooked.

The MICrONS dataset that contains layer 4 was collected from a P87 mouse, not a P36. While this error may not impact their overall conclusions. nevertheless, it does affect the figures and should be corrected.

We apologize for this oversight and confused the MICrONS L2/3 p36 age for also being the age of the MM³ dataset. We corrected all aspects of the manuscript including replotting the data, and all references to this age in the text.

Reviewer #3 (Remarks to the Author):

I had reviewed this paper in its previous life at Nature Neuroscience and found it to be a remarkably interesting and well executed set of studies. The authors have responded the the very few concerns that arose then. As a result I am happy to recommend acceptance.